# A Narrative Review about Nutritional Management and Prevention of Oral Mucositis in Haematology and Oncology Cancer Patients Undergoing Antineoplastic Treatments

**DOI:** 10.3390/nu13114075

**Published:** 2021-11-15

**Authors:** Balma García-Gozalbo, Luis Cabañas-Alite

**Affiliations:** Faculty of Pharmacy, Universitat de València, 46100 Valencia, Spain; balmagarciagozalbo@gmail.com

**Keywords:** cancer, oral mucositis, diet therapy

## Abstract

Cancer is a prevalent disease worldwide, and treatments such as radiotherapy and chemotherapy sometimes lead to adverse events. Oral mucositis is one of the most disabling adverse events, and clinical guidelines do not take into account nutritional interventions. The primary endpoint was to gather the evidence about the efficacy of nutritional interventions in the prevention and/or treatment of antineoplastic-induced oral mucositis in oncological patients. A bibliographic review was carried out in the PubMed data base by combining MeSH terms with Boolean operators. Articles were selected based on inclusion and exclusion criteria; 50 final articles were found. Although further evidence is needed, glutamine, honey, and vitamins appear to be good therapeutic options. The rest of the compounds presented controversial or insufficient results, making it difficult to draw conclusions over their utilization as prevention or treatment options. Little evidence is reported about oral mucositis nutritional interventions in spite of them being attainable and affordable compounds. Scarce evidence is shown in paediatric patients compared with adults. Developing higher quality studies and combinations with the compounds researched is necessary for creating a stronger body of evidence.

## 1. Introduction

The World Health Organization (WHO) conceptualizes cancer as a generic term which includes a wide group of diseases that can affect any part of the organism. They are also called “malignant tumours” or “malignant neoplasms”. A defining characteristic of cancer is the rapid division of abnormal cells that extend beyond the common limits and can invade adjacent parts of the body or spread to other organs, a process called metastasis. Metastasis is the main cause of death due to cancer disease [1]. It is indispensable to highlight the word “group” of diseases, considering that they are different entities with common characteristics, such as their genetic origin, uncontrolled proliferation, resistance to cell death, and capability of invading adjunctive tissues or metastasizing in distant organs [2]. Based on this definition, we can indicate the global situation on morbidity and mortality on account of cancer. From the data reported by the international agency for cancer research (IARC, Lyon) in 2018, we can elucidate that the incidence went up to 18.1 million cases, and the mortality climbed to 9.6 million. Additionally, lung, colorectal, and mammary cancers are still the most incident of all. Furthermore, death rates for women are lower than death rates for men [3]. When neoplastic processes are diagnosed, they come along with a treatment. Whether chemotherapy or radiotherapy is used, adverse events are found, with oral mucositis being one of the most common among others, such as diarrhoea or vomiting [4]. It is estimated that conditioning regimens for stem cell transplantation are the treatments that generate the highest rates of oral mucositis incidence, followed by radiotherapy and lastly by chemotherapy [5,6,7,8,9].

All things considered, it is now possible to define oral mucositis more specifically as the inflammation of the oral mucosa, with clinical consequences such as ulcers or erythema secondary to radiotherapeutic or chemotherapeutic treatments [10]. This term differs from the unit “mucositis”, which is considered to be damage and inflammation of oral, pharyngeal, laryngeal, and oesophageal mucosa, together with other areas of the gastrointestinal tract, secondary to antineoplastic treatments such as radiotherapy and chemotherapy [8]. Likewise, sometimes the word “stomatitis” is used as a synonym for oral mucositis (OM) even though it is a different entity because it defines the inflammation in the oral mucosa as being due to another specific aetiology that is unrelated to antineoplastic treatments (e.g., infections) [10,11].

Multiple scales are used to assess OM, but none of them have achieved agreement in the field as being the standard scale. The scales have been classified as general scales (e.g., WHO scale), scales with multiple variables (e.g., Beck, Eliers, and Walsh scales), and those that are specific to OM treatment (e.g., Oral Mucositis Rating, National Cancer Institute, Bethesda, and Radiation Therapy Oncology Group/European Organization for Research and Treatment of Cancer scales, Brussels) [12,13,14].

After OM is evaluated, it is possible to find either mild erythema and a burnt mouth sensation or profound mouth ulcers and the inability to eat. Independent of the clinical presentation, the cause of OM relies on a matrix of heterogeneous processes. There is not one physiopathological mechanism that is the direct cause of OM, but there are instead many complex biological routes. OM is typically based on five stages [8]. The first two of stages have an immediate appearance after chemotherapy, radiotherapy, or chemoradiotherapy. The first phase is the so-called “initiation” phase, and it is based on the death of basal epithelial cells (as a consequence of damage to the DNA in cells as a result of treatment) along with the secretion of reactive oxygen species (ROS) and endogenous damage-associated pattern molecules (CRAMPs), the last of which binds to specific receptors and sets in motion the second stage. This second stage is designated as the “primary damage response” and involves a cascade of biological events that interact with one another and that conclude by activating various transcription factors (nuclear factor kappa-B (NF-κB), Wnt, p53, and their associated canonical pathways) [8,10,11,12]. For example, NF-κB route activation can occur directly (by antineoplastic treatments) or indirectly (because of ROS and CRAMPs), showing evidence of the complexity of OM physiopathology. The activation of transcription factors produces the expression of a large number of genes, some of them being related to the production of molecules (e.g., COX-2, inducible NO-synthase, superoxide dismutase, etc.) that take part in the secondary emergence of OM. Other routes directly linked with the onset of this pathology exist, such as the nitrogen metabolism pathway, ceramide and fibrinolysis route, and the stimulation of matrix metalloproteinases (MMPs) [11,12].

The third stage is called “signal amplification” and occurs when the primary response molecules have positive or negative feedback on the local tissue. Throughout this period there is no visible injury even though the submucosal tissues and basal membrane are already damaged. Four to five days after antineoplastic treatment, the destructive processes of the three first stages triggers the fourth stage, called the “ulceration stage”. This phase implies the ulceration of the oral mucosa (transecting the full epithelial thickness), with patients being more prone to infections when this happens. Moreover, the ulcers are colonized by oral bacteria, which worsens the initial injury and makes it last longer due to infiltrating macrophages that generate pro-inflammatory cytokines. The “healing stage”, or the last phase, consists of the spontaneous remission of the oral cavity injuries. This event happens as a result of activation from signalling molecules (extracellular matrix) that direct the migration, proliferation, and differentiation of the epithelium bordering ulcerative areas [11,12].

Managing OM is complex. Clinical practice guidelines (CPG) on this topic are scarce and the vast majority are not recent, although there are exceptions. The Multinational Association of Supportive Care in Cancer and the International Society of Oral Oncology (MASCC/ISOO, Toronto, ON, Canada) have published the most recent clinical practice guideline on the topic of oral mucositis prevention and treatment.

Strategies such as basic oral care are beneficial practices, but they have a low grade of evidence. Therefore, procedures such as a multiagent combination of oral care protocols (providing guidance on the time, frequency, and products that patients with cancer should use every day) are considered beneficial for preventing OM from appearing, and other treatments such as chlorhexidine rinses are contraindicated for the prevention of OM (grade III evidence); nevertheless, saline or bicarbonate sodium rinses, patient education, and professional oral care have insufficient evidence with which to determine whether they positively or negatively impact OM. The use of benzydamine rinses is recommended to prevent OM in patients with head and neck cancers undergoing radiotherapy (RT) or chemotherapy (CT) (grade I and II evidence respectively); on the other hand, with level II evidence, photo-biomodulation (PBM) is recommended in patients with hematopoietic stem cells transplantation (HSCT) for the prevention of OM. Other anti-inflammatory drugs were studied and none of them had enough evidence to present a recommendation [8,15,16].

Regarding other drugs, sucralfate is not recommended as prevention or treatment, whereas topical morphine (0.2%) is suggested for OM treatment when it is associated with pain (low evidence grade III) in patients with head and neck cancer undergoing CT and RT. In order to prevent OM onset, oral cryotherapy (ice therapy) is recommended thirty minutes before patients receive 5-fluorouracil (5-FU) boluses when undergoing CT, or when a patient is treated with melphalan in high doses prior to an autologous stem cell transplantation (which means that the donor and receptor are the same human) [8,15,16].

In patients affected by haematologic cancer, the use of intravenous keratin growth factor (KGF-1) is recommended for the prevention of OM before an autologous stem cell transplantation with conditioning regimens including high dose CT and total body irradiation (TBI). In this context, neither topical granulocyte-macrophage colony-stimulating factor (GM-CSF) nor parenteral glutamine should be used in the prevention of OM in stem cell transplantation in contrast to what has been said about KFG-1. Oral glutamine plus disaccharide was shown to significantly help mucositis during chemotherapy and autologous BMT in randomized, placebo-controlled trials. However, experts suggest using oral glutamine to prevent OM from appearing in patients with head and neck cancer who are undergoing CT and RT (grade II evidence). Finally, the use of honey has been considered to prevent OM in patients with head and neck cancer who are treated with RT or CT and RT [15,16,17,18].

The existing clinical practice guidelines about OM do not address, or poorly address, a nutritional treatment approach, leaving unresolved doubts and evidence voids regarding which interventions should be implemented. This study intends to address which nutrients can be administered to prevent and/or treat CT- and RT-induced OM in cancer patients.

## 2. Materials and Methods

### 2.1. Literature Research

Main aim: To establish the effect of nutritional interventions for the prevention or treatment of oral mucositis in cancer patients undergoing radiotherapy and/or chemotherapy. This paper is based on a bibliographic review of information published from 2000 to 2021 and free full text. The literature chosen for review was published between October 2020 and April 2021 and includes the latest research results based on the search strategy used.

The bibliographic research was carried out using the PubMed and Medline databases, finding 252 results. Key search terms, including MeSH terms and related terms, were determined in accordance with the PICO method: mucositis, oral mucositis, mucositis and nutrition therapy, medical nutrition therapy, nutraceuticals, and dietary formulation and excluding force feeding or gastric feeding tubes, and related terms.

### 2.2. Study Eligibility, Selection, and Data Extraction

On the one hand, research articles were selected taking into account the following inclusion criteria: (1) original articles, (2) articles published between 2000 and 2021, and (3) those written in English and Spanish. On the other hand, exclusion criteria were: (1) reviews with or without meta-analysis, (2) article topics that were unrelated to the research subject matter, (3) articles whose aim was treating animals as a veterinary care strategy, and (4) qualitative research articles.

The preliminary screening of studies was completed by reviewing the titles and abstracts to assess eligibility. Studies that did not align with the inclusion and exclusion criteria at this point were excluded. Of the remaining studies, the full text articles were reviewed to determine inclusion in a second screening process, and this review process was completed by two reviewers independently and blinded—in other words, without knowing the other reviewer’s decisions, the scientific journal, the reference, or the author.

The extracted data included study design, subject characteristics (disease, treatment), intervention, and outcomes (incidence, weight loss, mortality).

## 3. Results

Following the literature search, 254 articles were identified. After excluding 204 of them, 50 articles that met the inclusion criteria remained and were included in the final review. Figure 1 further outlines the selection process used.

As shown in Figure 1, 46 articles were excluded after the first screening process conducted by reading the title and abstract of each article. Out of these 46 articles, 9 of them were excluded due to their language, 1 because of duplication, and 36 for being reviews. In this article, 208 references were subsequently selected for submission to a second screening process which consisted of a full reading and an analysis of its relevance for inclusion in the review. Those references that were not considered useful for this study were excluded (158).

A narrative analysis of the studies revealed two types of population (paediatric and adults), and three key themes emerged: glutamine, honey, and other dietary components or prevention methods. The results were classified according to these themes. Articles included in the final review can also be seen in Table 1.

### 3.1. Summary of the Studies Included

#### 3.1.1. Paediatric Population

Of the final articles included, six of them addressed the prevention and/or treatment of OM in paediatric patients with cancer [27,29,44,47,63,68]. The ages evaluated in those articles ranged from one to nineteen years old, varying in the different scientific articles. Additionally, all of the articles primarily addressed haematological neoplasms, even though some included solid tumours in their sample (in a lower proportion). Four of the studies were based on clinical trials, whereas another was based on a cohort study and the last was a case series study. The four clinical trials studied different compounds.

#### 3.1.2. Glutamine

From the reviewed articles, 18 articles retrieved were related to different preparations of glutamine for the prevention and treatment of OM in adults with cancer [19,20,21,22,23,25,27,28,32,33,35,38,40,41,46,48,49,54,56,58]. All of the studies were performed in a population that ranged from 19 to 70 years old. The most frequent age groups were those ranging from 30 to 50 years old. The overall articles consisted of 15 clinical trials [19,20,21,23,25,27,32,33,35,38,40,41,46,48,49,54,56], 1 cohort study [22], and 2 laboratory studies (1 in vivo [58] and the other in vitro and in vivo [28]).

Regarding clinical trials, it can be stated that their nature was heterogeneous. More specifically, heterogeneity was due to the fact that different levels of methodological blindness existed (double and triple blindness, open trials, preliminary, etc.) and because there were differences in the treated disease, the used therapy (CT, RT, or CRT), and even in the OM grading scale.

#### 3.1.3. Honey

Another compound studied as a possible prevention and/or treatment for OM was honey—having retrieved eight articles about it after preliminary research,. Out of those, seven articles were designed as research with different types of blinded clinical trials [36,43,49,52,55,56,59] and one of them was a laboratory study (in vitro and in vivo) [24].

#### 3.1.4. Other Dietary Components or Prevention Methods

Vitamins were another group that was analysed in different studies (seven articles found) [29,31,63,65,66,68].

Liquorice was studied in three clinical trials, each establishing different results [34,57,62]. Cryotherapy was examined in two articles in which a reduction in OM incidence and severity was determined [53,61].

It must be highlighted that there were also two compounds analysed (date palm pollen and polydeoxyribonucleotides) in the present study that were not possible to compare with the existent scientific literature since they were only researched in one article [51]. However, their results were positive, demonstrating the need for further evidence on the provision of new approaches to the prevention and treatment of OM induced by antineoplastic therapies.

## 4. Discussion

### 4.1. Paediatric Evidence

Al Jaouni et al. (2017) [47], through an open clinical trial, determined that honey diminished the incidence of the worst oral mucositis grades and delayed its onset. It was also found that honey helped to decrease pain and infections and to reduce the length of hospital stay. Furthermore, honey increased patients’ weight compared with the control group. In contrast, Widjaja et al. (2020) [27], with their double-blind randomized controlled trial showed a lower incidence of OM in their treatment group with glutamine, proving other results such as a decrease in the severity of OM due to the glutamine and a reduction in treatment duration and lower sanitary costs.

Alternatively, Thornley et al. (2004) [63] demonstrated an OM incidence and severity reduction along with lower regimen-related toxicity (especially in high-risk patients) thanks to a preparation made of ursodeoxycholic acid, vitamin E, folinic acid and administered as parenteral nutrition as prevention. In contrast, Pattanakitsakul et al. (2020) [68] proved in their preliminary quasi-randomized trial that vitamin A did not prevent OM.

Oosterom et al. (2019) [29] in their cohort study provided evidence that basal vitamin D levels were not related to MTX-induced OM, but an association was found (OR = 1.26) between the reduction in vitamin D levels during MTX treatment and severe OM (grade 3 or more). Finally, a case series (3) reported the benefits of honey as an OM treatment in patients undergoing chemotherapy [44]. Sung et al. [69] carried out a clinical practice guideline in which they collected all the available evidence about glutamine (and other treatments) for preventing OM in paediatric patients undergoing a conditioning regimen prior to HSCT, and they determined that there was no consistent reduction in OM in more than one study. The only retrieved study that was reviewed in this paper about glutamine in paediatric patients revealed a positive change, with the clinical trial presenting consistent evidence of the effectiveness of glutamine in decreasing the incidence of OM, highlighting the need for more studies before recommending the administration of glutamine (Widjaja et al., 2020) [27].

In addition, Sung et al. (2015) [69] established in their review that there was a study of a topical vitamin E treatment that did not demonstrate an effective reduction in OM. In the present study, it was not possible to compare the retrieved article with the Sung et al. (2015) [69] review since it applied a combination of agents. Nevertheless, it must be emphasised that because of the huge lack of studies with vitamins, it is necessary to identify their effect and implement recommendations based on sufficient data (Thornley et al., 2004) [63].

The cohort study with vitamin D was not possible to compare with others because additional papers did not exist, corroborating the assertion in the previous paragraph about the lack of evidence on this subject (Oosterom et al., 2019) [29].

Friend et al., (2018) [70] clarified in their review that honey could be effective in the treatment and prevention of OM in paediatric patients in limited resources areas but determined that there was no available evidence that compared whether honey was as effective or more effective than other established treatments. The evidence found in this study supported this assertion since the open clinical trial showed the efficacy of honey in OM, but there were no comparative studies between honey and other treatments. Moreover, case series do not have strong enough evidence to be able to reasonably compare them with clinical trials or to state conclusions if additional evidence does not exist (Al Jaouni et al., 2017) [47].

### 4.2. Glutamine

Regarding adults and the use of glutamine, Nihei et al. (2018) [40], Chattopadhyay et al. (2014) [41], and López-Vaquero et al. (2017) [48] in their scientific studies did not show a reduction in the total incidence of OM. In contrast, Huang et al. (2019) [19] and Tanaka et al. (2016) [21] showed evidence of a reduction in the total incidence of OM; this matches the contribution of a cohort study made by Pachón-Ibañez et al. (2018) [22], except for the fact that they did not report lower incidence in the severity of OM (grade ≥ 2). The trials conducted by Huang et al. (2019) [19] and López-Vaquero et al. (2017) [48] obtained the same result on this matter, contrasting with the results of Tsujimoto et al. (2015) [20], Tanaka et al. (2016) [21], Nihei et al. [40], and Chattopadhyay et al. (2014) [41] who observed less severity of OM in patients treated with glutamine.

Concerning the time of onset of OM, only three papers addressed this topic, and two of them determined an absence of delay in the onset of OM, contrary to what the Chattopadhyay et al. (2014) [41] study addressed [20,48]. As for the duration of OM, it was studied in four trials, with a reduction in the duration of the most severe OM proven in two of them [41,64].

Other nutritional factors, such as weight, were studied in five scientific articles, showing a slightly minor tendency in weight loss in those groups with glutamine [21,23,25,33,64]. Additionally, Tanaka et al. (2016) [21] proved that weight of cancer patients could be maintained by a mixed glutamine and elemental diet [23,25,33,64].

Quality of life (QoL) was evaluated by López-Vaquero et al. (2017) [48] through an adapted questionnaire for this topic, but without observing a link between the use of glutamine and OM patients’ quality of life. Other factors such as pain and the use of analgesia (three articles) [20,22,51], dysphagia (one article) [55], and odynophagia (one article) [55] were studied. The results provided determined that treatment with glutamine improved dysphagia, odynophagia, and pain, showing in two of the studies a pain decrease with analgesia use too [20,22,33,40,55].

Four of the trials delved into a product called Elental ^®^ [28,32,35,38], a liquid dietetic formula enriched with amino acids and a source of L-glutamine. The results showed at large a reduction in the severity of OM in groups treated with Elental ^®^, highlighting the investigation of Harada et al. (2019) [35] that also showed a decrease in the administered analgesia and CRP levels (4–6 weeks of CRT). These results concurred with those obtained by Harada et al. (2018) [28] in their in vivo study in which rats treated with Elental ^®^ healed more rapidly from OM ulcers [32,38,46].

Regarding its combination with other products, Peterson et al. (2007) [54] analysed a compound called Saforis ^®^ (an oral formulation that increases the availability of glutamine in the oral cavity), which proved to decrease the severity of antineoplastic-induced OM. Even though not currently available, a commercially available glutamine + trehalose powder for use as a suspension is now accessible [71]. Finally, Bateman et al. (2013) [58] used a combination of whey protein, fatty acid, and glutamine, showing that the combination does not protect against OM in rats. Nevertheless, Anderson and Lalla (2020) [71] performed a review in which they suggested glutamine rinses (liquid formula) in patients with head and neck cancers undergoing CRT since they found that the rinses decrease the severity and duration of OM and esophagitis. This is comparable with five of the found studies in the bibliographic search, even though all of them studied oral glutamine for rinsing and swallowing afterwards, revealing the same benefits reported in the commented review. Oral glutamine might be beneficial by topical absorption or ingestion if local uptake can be facilitated with a disaccharide. Although other studies have also shown modest benefits for glutamine rinses or ingestion, it would appear that achieving a local effect may possibly increase effectiveness [19,20,22,41,48,71].

In addition, Shuai et al. (2020) [72] recently published a meta-analysis affirming that oral glutamine had no clinical benefits in the prevention and/or treatment of CRT- or RT-induced OM in patients with head and neck cancers. Therefore, glutamine should be studied further in order to obtain more consistent results.

### 4.3. Honey

Severity is the most studied outcome of honey use. It has been proven that honey is effective in the eventual reduction in the severity of OM, as Howlader et al. (2019) [49], Rao et al. (2017) [36], and Amanat et al. (2017) [43] point out in their scientific publications.

Along the same lines, Raeessi et al. (2014) [52] indicated that the efficacy of the combination of honey and coffee for decreasing the severity of OM exceeded the same capacity of honey by itself and steroids. On the other hand, Fogh et al. (2016) [55] delved into the severity of esophagitis, and they did not find a link between honey and the decrease in OM stage. This same study did not highlight an association between honey and a reduction in late forms of odynophagia (it does not occur in the same way in precocious forms).

Rao et al. (2017) [36] approached the incidence of OM and proved the benefits of honey in the prevention and delay of the onset of OM. In addition, they showed a reduction in weight loss in their treatment group with this compound. Jayachandran and Balaji (2012) [59] also observed a delay in the onset of OM in patients treated with honey in comparison with those who were treated with benzydamine; in fact, no effects were found on the duration.

Regarding other parameters analysed, only the study by Howlader et al. (2019) [49] assessed the influence of honey on quality of life (QoL) during OM, observing a late improvement in OM because of the administration of honey. The trial performed by Samdariya et al. (2015) [56] showed a consistent decrease in the severity of pain, resulting in fewer discontinuances of RT.

As a final remark, Anturlikar et al. (2019) [24] in their in vitro and in vivo study with a combination of turmeric, Triphala, and honey (HTOR-091516) showed positive results such as low product toxicity, inflammation inhibition (TNF-α), and a protective effect against CT-induced OM.

Münstedt and Männle (2019) [73] in their review evaluating the use of honey as an OM treatment showed a decrease in the severity of the ulcers, which harmonized with the scientific articles found in the present paper and in the systematic review and meta-analysis conducted by Tian et al. (2020) [74].

Other aspects mentioned by Tian et al. (2020) [74] include the indication of a potential decrease in the incidence of OM during the treatment with honey; however, from the present study, only one study proved significant prevention.

There exist other interesting findings, such as OM pain reduction and the inhibition of the inflammatory pathways (in vitro study with “HTOR-091516”), that are not comparable with the mentioned reviews since there is no evidence to suggest or recommend the use of honey under these circumstances. Even though honey could be useful for reducing mucositis severity, the rest of its uses should be studied more in-depth to provide reliable data and to determine suggestions and recommendations on this matter.

### 4.4. Vitamins and Amino Acids

The most important results of the three retrieved clinical trials showed a reduction in the severity of OM in patients treated with a combination of GeneTime© [30], group B vitamin complex, Oncoxin© [26] (combination of vitamin C, B6, and amino acids), and vitamin B9 [37], proving a reduction in mucositis incidence too. Nevertheless, Branda et al. (2004) [37] in their cohort study did not find a link between the quantity of vitamins B12, B9, and multivitamins supplements and OM severity.

Specifically, Oncoxin© was related to a weight increase and a normal food intake too [26]. On the one hand, the combination of GeneTime© and vitamin B proved to result in less pain and quicker healing of OM ulcers [30]. Finally, treatment with vitamin B9 coincided with the effect of Oncoxin© in the improvement in food intake capacity, but it did not show a reduction in the amount of time using parenteral nutrition; however, it did show a reduction in opioid use [26,30,60].

On the other hand, Nejatinamini et al. (2018) [31] showed that the decrease in the blood levels of vitamins A and D during RT were related to OM. In the same way, an in vitro study determined relevant information since it showed that vitamin E (γ-tocotrienol) in CT fostered the survival of oral human keratinocytes through the inhibition of reactive oxygen species (ROS), probably by the suppression of the Nrf2 route [65].

Recently, a clinical trial published by Agha-Hosseini et al. (2021) [66] with a rinse made of vitamin E, hyaluronic acid, and triamcinolone proved to be useful in the treatment of OM. Similarly, another scientific group did a clinical trial with folinic acid for the prevention of OM. However, their results did not prove to be very successful [67].

Related to the use of complete protein, the study of Perrone et al. (2017) [45] proved that there was not a link between the intake of concentrated whey protein and the incidence, severity, and duration of OM. However, the individuals who took a higher quantity of whey protein presented less severity and duration of OM compared with those who took a lower quantity of protein. Finally, an in vivo study by De Sousa et al. (2018) [39] indicated that glycine promoted major and better tissue restructuring.

Regarding non-combined amino acids, a clinical trial and an in vivo laboratory study were highlighted. The rest of the studies related to amino acids (three in total) appeared in combination with other nutritional compounds and have already been mentioned.

Yarom et al. (2019) [75] considered that vitamin B9 could not be effective in the prevention of OM in patients who received therapy for a posterior HSCT, highlighting a cohort study (which was also analysed in this study) that proved efficacy in the reduction of both total incidence of OM and OM severity in patients undergoing CT for a posterior stem cell transplantation. There exist few studies related to vitamin B9 that have attempted to seek evidence about its benefit.

An evidence void also occurs when studying the treatment of OM with vitamins E and D since both have some studies (four about vitamin E [31,63,65,66], one about vitamin A [68], and one about vitamin D [29]) that determined different effects of the vitamins on OM and thus cannot be compared due to the characteristics used to analyse each study (they used different laboratories, cohorts, clinical trials, etc.). Therefore, supporting what Yarom et al. (2019) [75] mention in their review, it is not possible to determine consistent evidence for the use of vitamins in the treatment of OM.

In this section, when it comes to analysing the use of amino acids, the fact that there are no reviews about the use of glycine and concentrated WHEY protein was highlighted. As a result, there was a lack of evidence found in this study (two articles, one on each of the topics) for determining whether amino acids could be useful in treating OM.

### 4.5. Glycyrrhiza Glabra

On the one hand, the study of Mamgain et al. (2020) [34] and Das et al. (2011) [62] found both a reduction in the incidence and severity in those groups treated with liquorice (Glycyrrhiza glabra), with more effect than those treated with honey in the same trial. Contrastingly, Matsuda et al. (2015) [57] proved that the use of Hangeshashinto (mixed with seven medicinal plants, Glycyrrhiza glabra among them) did not affect the incidence and stage of OM, but it did affect the duration of severe OM.

Finally, it should be highlighted that Das et al. (2011) [62] also found fewer interruptions in the treatment and persistence of xerostomia despite liquorice administration.

OM therapy with liquorice is novel and not often researched. In the present study, there were only a few articles analysed about this compound, making it insufficient for determining clear evidence in support of the compound’s preventive and/or healing effect in OM. Richard (2021) [76] proved in a review that glycyrrhizin and glycyrrhetinic acids have anti-inflammatory properties, specifically inhibiting interleukin secretion (IL-6, IL-2, IL-12, etc.), the expression of TNF-α, and cytokine cascade, among others.

Even though this literature review did not focus on OM specifically, attention could be drawn towards this matter, since physiopathologically many inflammatory routes are involved in the production of oral ulcers, and compounds such as liquorice (with anti-inflammatory properties) could give results showing the inhibition of inflammatory routes if further research is carried out in the future.

### 4.6. Others

Elkerm and Tawashi (2014) [51] delved into the effect of date pal pollen in OM treatment through a preliminary study. Such research showed a reduction in the incidence and severity of RT-induced OM. A reduction in the pain and dysphagia during OM was also proven.

Second, deoxyribonucleic acid was studied by Podlesko et al. (2018) [44], using it as a topical spray in a case series. The results showed pain relief and quicker healing of OM ulcers in two cases and a deterioration in a third case.

López-González et al. (2021) [77] proved the efficacy of oral cryotherapy for preventing and treating OM. This evidence was also found in several clinical practice guidelines, highlighting its best utility in the prevention of OM in patients treated with CT. The present study cannot compare the results of the given evidence due to the scarceness of the analysis (two articles [53,61]), but it does agree with the results found in the evidence retrieved from other reviews and clinical practice guidelines [8,15,16].

### 4.7. Strengths

The implementation of this paper allowed the gathering of information related to a rarely studied topic, opening at the same time future doors to investigations about this subject and providing evidence of the need for more scientific literature about this pathological entity that is so prevalent among oncological patients.

In the same way, carrying out this review helped to identify the best available evidence on OM nutritional treatment (either prophylactic or as a treatment in and of itself). Likewise, the review provides a guide for health professionals involved in the patient care process, such as dietitian–nutritionists, nurses, and doctors.

### 4.8. Limitations

Some methodologic limitations of the present study must be highlighted. On the one hand, the available time to carry out this review was highly limited (from October to February) and only included studies in English and Spanish languages that were available in one database (PubMed), affecting the extent of the bibliographic search; this selection bias may have an impact on the results and should be considered in systematic reviews which should include results in different languages. On the other hand, a context limitation was present since the lack of consensus in the use of OM grading scales in the different scientific articles was noted, making it impossible to compare findings between the articles even though they assessed the same nutritional compounds.

Lastly, nutritional interventions in this disease entity are not studied as much as they should be studied; therefore, there is not enough scientific literature to affirm solid conclusions, requiring more support with further research articles.

## 5. Conclusions

In paediatric ages, positive results were found for the treatment and/or prevention of OM, but more investigation is needed since an evidence void exists when it comes to the use of all of the studied nutritional compounds.

Regarding the adult population, glutamine and honey could be the most useful treatment for OM, but more evidence is needed to confirm a reduction in OM severity. In vitamins and amino acids (different from glutamine), little evidence exists. It seems like vitamins could serve as a treatment, but there is scarce evidence, and generally, the investigated formulations did not only include vitamins, making it difficult to prove vitamins’ effects on their own.

It was not possible to present the usefulness of Glycyrrhiza glabra, date palm pollen, and polydeoxyribonucleotides in the prevention and/or treatment of OM in oncological patients due to a lack of evidence. Moreover, cryotherapy was also analysed insufficiently in this study, contrasting with what was found in the scientific literature.

In general, very few scientific publications exist on the nutritional approach to CT- and/or RT-induced OM in cancer patients, even though there is great interest in the study of alternative OM treatments and such studies are highly attainable and easily available.

## Figures and Tables

**Figure 1 nutrients-13-04075-f001:**
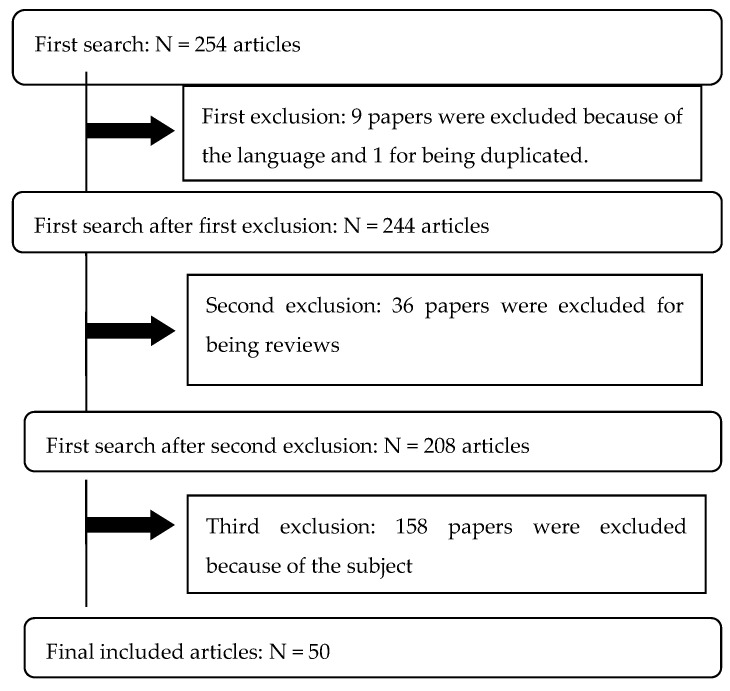
Data Collection for the study.

**Table 1 nutrients-13-04075-t001:** Summary of studies investigating prevention or treatment of Oral Mucositis in cancer patients with dietary components or prevention methods.

Author (Year)	*N*	Study Design	Objective	Intervention	Time (Months)	Conclusions
Huang et al. (2019) [19]	59	RCT Phase III, double-blind	To evaluate whether oral glutamine prevents acute toxicities (OM and dermatitis) secondary to the treatment with RT in patients with head and neck cancers.	Glutamine (TG); Control, maltodextrin (CG); NCI CTCAE 4.03 version.	19	CG developed more OM (grades 2–4) than TG; nevertheless, the efficacy of the treatment with oral glutamine was not meaningful after RT in head and neck cancers.
Tsujimoto et al. (2015) [20]	40	RCT, double-blind	To evaluate whether oral glutamine reduces mucositis severity induced by CRT in patients with head and neck cancer.	Glutamine (TG); Placebo (PG); NCI CTCAE 3.0 version.	36	Oral glutamine reduced OM severity produced by CT in head and neck cancer patients with a maximum mean grade of OM lower for TG than for PG, a duration without meaningful difference between both groups, and a shorter duration of artificial nutrition required in TG.
Tanaka et al. (2016) [21]	30	RCT Phase II	To assess whether glutamine and the combination of glutamine and elemental diet reduce the incidence of CT induced OM in patients with oesophageal cancers.	Placebo (PG); Glutamine (TG); Glutamine + elemental diet (CTG); NCI CTCAE 3.0 version.	36	No differences between PG and TG
Pachón-Ibañez et al. (2018) [22]	262	Prospective cohort study	To evaluate whether oral glutamine prevents mucositis induced by oncological therapies (CRT or RT) in patients with head and neck cancer.	Oral glutamine (TG); Placebo (PG); RTOG/EORTC (MO and oesophagitis).	-	More OM in PG (RR:1.78), without remarkable differences in severity. Higher odynophagia in PG (RR:2.87), with more severity (R = 4.33). In PG, more discontinuity in the treatment.
Chang et al. (2019) [23]	60	RCT Triple-blind	To measure the impact of oral glutamine as a supplement for the prevention of oesophagitis induced by CRT in patients with advanced non-small cell lung cancer (stages III–IV).	Placebo (PG); Oral glutamine (TG); ARIE, acute radiation induced oesophagitis.	12	TG had less severe ARIE than PG, as well as a reduction in the incidence of weight loss in TG.
Anturlikar et al. (2019) [24]	20	In vitro and in vivo study	To measure the safety and efficacy of “HTOR-091516” (Tumeric, Triphala, and honey) as a treatment for OM induced by 5-FU (CT).	IN VITRO: gingival human fibroblast, mouse connective tissue, and human oral reconstructed epidermis culture. Later treatment with “HTOR-091516” and MTT test (cytotoxicity) + TNF-α inhibition test (inflammation)+ test INVITTOX SKINETHIC ™ (irritation). IN VIVO: two groups of rats (control (PG) and treatment with “HTOR-091516” (TG)). In TG: drops on the induced ulcer in every animal during their treatment with CT.	0.5 (14 days)	The average weight loss in TG was lower; there also was less mortality and a reduction in OM grade (WHO scale). The product is proposed to prevent OM.
Cho et al. (2019) [25]	91	CT	To assess the effect of glutamine-enriched parenteral nutrition (PN) on weight, infections, complications (mucositis, neutropenia, and graft-versus-host disease), and mortality in patients who underwent HSCT transplantation.	Control (CG); Glutamine (TG) through Dipeptiven© (bottle).	48	No significant association in the case of OM duration. A reduction in 100-days mortality for TG was noted.
Shumsky et al. (2019) [26]	15	Pilot RCT	To evaluate the efficacy of Oncoxin (ONCX) in oncologic patients with OM who underwent CT, RT or both.	Control group (CG); Group ONCX (OG); WHO Scale.	0.6 (20 days)	Lower-grade OM was found in OG (after 7 days of treatment and towards the end of their treatment).
Widjaja et al. (2020) [27]	48	Double-blind RCT	To measure whether oral glutamine prevents OM during CT (methotrexate (MTX) in paediatric patients with acute lymphoblastic leukaemia (ALL).	Placebo group (PG); Oral glutamine group (TG); WHO Scale.	0.5 (14 days)	There was a reduction in the incidence and severity of OM in TG after CT.
Harada et al. (2018) [28]	50	In vitro and in vivo study	To appraise the efficacy of Elental© (dietetic liquid formulation enriched with amino acids, which is a source of L-glutamine) in 5-FU (CT)-induced OM and dermatitis treatment.	In vivo: Saline solution (PG); Dextrin group (DG); Elental group (EG).	0.25 (8 days)	In vivo, OM healed faster than in EG.
Oosterom et al. (2019) [29]	99 (A) and 81 (B)	Cohort study	(1) To study the prevalence of vitamin D deficiency in paediatric patients; (2) To establish a link between vitamin D levels and methotrexate (MTX)-induced OM in paediatric acute lymphoblastic leukaemia (ALL).	Vitamin D levels before and after MTX therapy. After MTX, classification in two groups depending on OM grade: ≥3 or ≤3. NCI CTCAE 3.0 version.	2004–2012	There was no association between basal vitamin D levels and MTX-induced OM, but low levels of vitamin D during MTX therapy were found to be related to severe OM.
Sun et al. (2019) [30]	100	Double-blind RCT	To examine the effects of a group B multivitamin complex combined with GeneTime© (human recombinant growth factor) on the treatment of OM in patients with head and neck cancer undergoing RT.	Control group with vitamin B complex.Observational group with (OG) vitamin B complex + GeneTime©. RTOG scale.	12	Less severity, affected area, and healing time of the OM ulcers in OG.
Nejatinamini et al. (2018) [31]	28	Cohort study	To evaluate the changes in vitamin status during the treatment of head and neck cancers related to body composition, inflammation, and mucositis.	Dietetic intake measurement (3 days). Vitamin levels (vitamins D, E, B9, and B12) basal (before treatment) and 6 to 8 weeks after RT treatment (with or without CT); C-reactive protein (CRP) measurement. C-reactive protein (CRP) baseline before and after 6 to 8 weeks of RT treatment, either with or without QT after treatment.	1–1.5	Higher rates of OM were observed related to less vitamin D, B12, E, and B9 intake and lower blood levels of vitamin A and D.
Tanaka et al. (2018) [32]	19	RCT	To measure the intake of Elental© during two cycles of CT and to determine the incidence of OM in patients with oesophageal cancers treated with CT who completed their intake and those who did not completed it.	Elental© group (CG); Elental© group with uncompleted treatment (UG); CTCAE 3.0 version.	2 (56 days)	Less severity of OM in CG during CT with the use of Elental©.
Pathak et al. (2019) [33]	56	RCT	To assess the efficacy and role of oral glutamine in the treatment of OM and dysphagia induced by chemoradiotherapy (CRT) in patients with oropharynx and larynx carcinoma.	Control group (CG); Glutamine group (TG); NCI CTCAE 4.03 version.	1.75 (49 days)	TG had fewer hospitalizations due to OM and dysphagia. More incidence and severity of OM in CG.
Mamgain et al. (2020) [34]	127	RCT	To evaluate the efficacy of an ayurvedic preparation (based on *Glycyrrhiza glabra*) in order to decrease the severity of mucositis in patients with head and neck cancers who received chemoradiotherapy (CRT).	Comparison between basal and post-RT characteristics: (1) Conventional OM treatment (antiacids and anaesthetics) (CTG); (2) Conventional treatment and ayurvedic preparation (ATG); (3) Honey and conventional treatment (HTG).	24	Less severity, less pain, and shorter onset time of mucositis, both in ATG and HTG, but especially in ATG.
Harada et al. (2019) [35]	50	Open RCT	To evaluate changes in OM (injuries’ size, pain, and redness + CRP in plasma) in patients with oral squamous cell carcinoma undergoing CRT or RT with Elental© administration.	Elental group (EG); Non-Elental group as the control (CG); CTCAE 4.0 version.	24	In EG, milder OM development in CRT; no difference in RT.
Rao et al. (2017) [36]	49	Blinded single-centre RCT	To evaluate whether honey causes interference with RT-induced tumoral response or whether it exhibits a positive effect against OM in patients with head and neck cancer.	Povidone–iodine group (CG); Honey group (HG); RTOG scale.	6	Lower incidence of OM and less severity in HG. The implications for treatment interruption were not significant.
Branda et al. (2004) [37]	68	Pilot cohort study	To study the influence of B12 vitamin, folate, and dietetic supplements on CT-induced toxicity in breast cancer patients.	Blood samples (B12, B9, and neutrophils) before/after the first CT cycle; Questionnaires about supplement usage; OM grading (author-modified CTCAE); 68 subjects + historical controls.	-	No evidence of influence was found.
Okada et al. (2017) [38]	20	Pilot single-centre RCT	To evaluate the influence of Elental© on CT-induced OM and diarrhoeas in patients with oesophageal cancer.	IG: use of Elental©; CG: no Elental used; Questionnaires and clinical examination; CTCAE 4.0 version.	0.5 (14 days)	Less severe OM incidence in IG.
De Sousa et al. (2018) [39]	40	In vivo study	To evaluate the effects of glycine in the expression of collagen and platelet and epidermal growth factors (PDGF, EGF) in an OM murine model.	Control group (CG); Intervention group (IG: glycine supplementation); Measurement of collagen percentage and type and EGF and PDGF percentage.	-	Positive effects in IG, with a better recovering rate (collagen increase and growth factors reduction).
Nihei et al. (2018) [40]	67	Single-centre RCT	To evaluate the efficacy and safety of L-Glutamine sodium azulene sulphonate in the treatment of CT-induced OM in patients with colorectal and breast cancer.	Intervention group (IG); Control group (CG: standard oral hygiene); CTCAE 4.0 version; NRS pain scale.	24	Lesser OM severity in IG. No significant differences were found regarding incidence.
Chattopadhyay et al. (2014) [41]	70	Single-centre RCT	To evaluate the influence of oral glutamine on RT-induced OM in patients with head and neck cancer.	Intervention group with oral glutamine (IG); No placebo control group (CG); WHO scale.	8	Lower incidence, severity, and duration of RT-induced OM in IG.
Üçüncü et al. (2006) [42]	35	Laboratory CT (rats)	To determine the preventive effect of Vitamin E (VE) and L-Carnitine (LC), alone or in combination, on OM and myelosuppression by RT.	5 groups: (1) No RT (control: saline + simulated radiation); (2) RT; (3) RT + VE; (4) RT + LC; (5) RT + VE + LC. OM measurement scale: Parkins et al.; Clinical/histopathological OM was evaluated; Levels of SOD and CAT (antioxidants) and MDA (oxidative stress indicator).	Follow-up: 4 days pre-RT—10 days post-RT.	VE and LC proved to be radioprotective agents on their own and not combined together, with lower severity and longer time to histological appearance of OM. Good tolerance and no adverse effects.
Amanat et al. (2017) [43]	82	Single-centre RCT	To assess the effect of honey on clinical grades of OM.	Honey group (HG); Control saline group (CG); RTOG scale.	12	Lower incidence and severity of OM in HG during the RT.
Podlesko et al. (2018) [44]	3	Case series	To evaluate the effects of topical application of deoxyribonucleic acid on three OM (moderate–severe) cases in patients with head and neck cancer.	Oral spray of polydeoxyribonucleotide (PDRN) as treatment.	1	Increased relief and remission of OM over time, without interruption in treatment or opioid intake.
Perrone et al. (2017) [45]	73	CT	To analyse the influence of dietary supplementation with whey protein concentrate (WPC) on the incidence of OM in patients undergoing HSCT.	WPC group (WG); Historical controls (CG); WG was sub-stratified into: consumed <80% PWC (WG1) or ≥80% (WG2) of the offered dose; WHO and CTCAE 4.0 version.	Not specified	No significant differences between WG and CG in incidence, duration, and severity of OM concerns. However, in WG2, shorter duration and lower incidence of severe OM was found.
Ogata et al. (2017) [46]	22	Pilot prospective study	To evaluate the preventive effects of Elental ^®^ on CT-induced OM in patients with colorectal cancer (CRC).	22 patients in Elental (1 group); CTCAE 3.0 version.	36	Significantly reduced CT (5-FU)-induced OM grade.
Al Jouni et al. (2017) [47]	40	Open RCT	To evaluate the effects of honey on grade 3–4 OM, reduction in bacterial/fungal infections, duration of OM episodes, and body weight in paediatric leukaemia patients undergoing CT or RT.	Control group (CG) with Lidocaine, Mycostatin, Daktarin, and oral cleaning; Experimental group (HG) with same routine as CG + honey (4–6 times/day); WHO scale.	12	Significant reduction in severity and pain in HG. Significant improvement in weight and time to OM onset in HG.
Lopez-Vaquero et al. (2017) [48]	49	Phase II double-blind RCT	To evaluate whether glutamine is effective in reducing the incidence and severity of mucositis and dermatitis induced by RT or CRT in patients with head and neck cancer.	L-Glutamine group (TG); Placebo group with malto-dextrine (PG); CTCAE 3.0 version.	6	Incidence and severity with no significant differences between groups.
Howlader et al. (2019) [49]	40	RCT (single-blinded).	To assess whether honey improves mucositis injuries and the quality of life of patients with RT/CT-induced OM (for head and neck cancer).	Treatment group (HG) with honey (both rinsed and ingested honey); Control group (CG) with saline solution; CTCAE.	From CT start—4 weeks after RT.	Less OM and associated symptoms induced by RT in HG. Shorter time towards the recovery of a regular quality of life.
Elsass (2017) [50]	10 (3 analysed)	Case series	The aim was to improve OM (oral comfort and feeding) with standard oral care and the use of Leptospermum honey in paediatric oncology patients after proven CT.	Application of honey on the buccal surface with a cotton swab, 3 times/day, Then spat or suctioned out.	-	Shorter healing time with lower pain rate in all cases.
Elkerm and Tawashi (2014) [51]	20	Pilot study	To evaluate whether date palm pollen (DPP) can be effective in the prevention and treatment of RT- and CT-induced OM in patients with head and neck cancer.	DPP group (one daily suspension); Control group (CG) (antifungal, rebamipide, and oral analgesia); OMAS score and visual analogue scale for mouth pain and dysphagia.	1.5 (6 weeks)	Significant reduction in incidence, severity, and pain in OM and dysphagia in DPP.
Raeessi et al. (2014) [52]	61	Single-centre double-blind RCT	To evaluate the effects of coffee + honey in the treatment of OM by CT and compare them with the effects of steroids.	3 groups: (1) Betamethasone group (EG); (2) Honey group (HG); (3) Honey + coffee group (HCG); WHO scale.	36 2011–2013	Significant reduction in the severity of OM in all three groups.
Baydar et al. (2005) [53]	99	CT	To research the effects of local cryotherapy on the prevention of CT (5-FU)-induced OM.	Intervention group (IG): CT courses with local cryotherapy (ice in the mouth during the CT course up to 10 min afterwards); Control group (CG): no-cryotherapy courses.	Not specified.	5-FU-induced OM incidence lower in GI (OR = 11.5).
Peterson et al. (2007) [54]	305	Phase III, double-blind RCT	To observe the efficacy of Saforis ^®^ in the prevention and treatment of OM caused by CT treatment in breast cancer.	Saforis group (SG); Placebo group (PG) (with subsequent cross-linking); WHO scale (ulcer grading scale); OMAS scale (ulceration measurement).	Not specified.	Lower severity and incidence rate in SG.
Fogh et al. (2016) [55]	119	Multicentric phase II RCT	To evaluate the effect of Manuka honey (liquid and tablets) in the prevention of RT-induced oesophagitis in lung cancer patients.	G1: Manuka honey (liquid); G2: Manuka honey (tablets); G3: control, standard care. Measurements: odynophagia (NRPS scale), pain, opioid use, dysphagia, weight loss, quality of life, and nutritional status.	12	There were no significant differences for groups G1, G2, or G3, so the use of honey did not prove to be superior to standard health care.
Samdariya et al. (2015) [56]	69	Open RCT	To study the intake of honey in pain relief caused by RT-induced OM in patients with head and neck cancer.	(1) Gargle with soda and benzidamine (PG); (2) Gargle with soda + benzidamine + honey (HG).	November 2011–January 2013.	Slightly greater relief in HG during the entire follow-up (3 months), with a significant reduction in the severity of OM-associated pain and fewer treatment interruptions.
Matsuda et al. (2015) [57]	90	Double-blind phase III RCT.	To research whether TJ-14 (Hangeshashinto) prevents and/or controls CT-induced OM in patients with colorectal cancer.	TJ-14 treatment group (IG); Placebo group (PG); WHO scale.	0.5 (14 days).	Significant reduction in the duration of severe OM, with no effect on the severity or incidence of OM itself.
Bateman et al. (2013) [58]	Not specified (approximate number = 48).	Laboratory in vivo study	To investigate the protective effects of nutritional drinks on the development of methotrexate (MTX)-induced gastrointestinal mucositis in animals with and without cancer.	G1: ClinutrenProtect ^®^ (whey protein, short-chain fatty acids, TGF-b, L-glutamine); G2: IMPACT AdvancedRecovery ^®^ (whey protein, medium chain fatty acids, arginine, nucleotides, and polyunsaturated fatty acids); G3: placebo (drink); Control: standard rat diet.	Not specified.	Administered diets offered no observed protection against MTX-induced mucositis.
Jayachandran and Balaji (2012) [59]	60	RCT	To evaluate the effect of natural honey and benzidamine hydrochloride on the development and severity of RT-associated OM in patients with oral cancers.	3 groups (oral rinses): (1) Honey (HG); (2) Benzidamine hydrochloride (BG); (3) Saline 0.9% (control group, CG).	6	Lower severity and earlier healing of OM not significant in HG.
Sugita et al. (2012) [60]	118	Retrospective CT	To ascertain the effects of folinic acid administration (systemic and rinsed) on the incidence of OM and acute graft-versus-host disease (GVHD) after GVHD prophylaxis with MTX in patients undergoing HSCT.	Systemic folic acid was administered to patients at increased risk of developing OM (*n* = 29).	48	Folinic acid could be useful in reducing the incidence of severe OM, both in systemic use and rinsed.
Sorensen et al. (2008) [61]	206	Double-blind RCT	To evaluate prevention of OM using chlorhexidine compared with cryotherapy during 5-FU CT in gastrointestinal cancer.	3 groups: (1) Chlorhexidine rinse (IG); (2) Placebo rinse (saline) (PG); (3) Cryotherapy (CG).	-	Higher severity rate of OM in PG and lower in CG, with a lower incidence and duration in IG and CG than in PG; therefore, prophylaxis seems effective in both IG and CG.
Das et al. (2011) [62]	52	RCT	Observe protective/healing effect against RT and CT effects (OM, skin reaction, xerostomia, or voice changes) when using *Glycyrrhiza glabra*.	4 groups: (1) Glycyrrhiza + local honey and oral Glycyrrhiza (GLHG); (2) Glycyrrhiza + local honey (LHG); (3) Topical honey (HG); (4) Control (conventional modern medication) (CG).	1.75 (7 weeks)	Lower incidence and severity in GLHG and LHG compared with CG, but similar to HG.
Thornley et al. (2004) [63]	37	CT	To determine the feasibility and potential efficacy of a fixed combination of agents in reducing RRT (regimen-related toxicity) in children undergoing HSCT.	Combination group of ursodeoxycholic acid (UDCA), vitamin E, folinic acid, and titrated parenteral nutrition (IG); Historical control group (1995–2000) (PG).	36	Significant decrease in the incidence and severity of OM in IG.
Iyama et al. (2014) [64]	44	RCT	To research whether supplementation with GFO (glutamine, fibre, oligosaccharides) decreases the severity of mucosal injury post-HSCT.	GFO group (IG); Control (CG); CTCAE 4.0 version.	36	Significant decrease in OM and higher survival rate in IG.
Takano et al. (2015) [65]	96-well plates	In vitro study	To investigate whether γ-tocotrienol (vitamin E) can enhance survival of oral human keratinocytes (RT7) against 5-FU-induced cell toxicity.	RT7 cells were treated with 5-FU and γ-tocotrienol. 4 groups: (1) γ-tocotrienol; (2) 5-FU; (3) γ-tocotrienol + 5-FU; (4) Control de 5-FU + N-acetylcysteine.	-	In C there was a significant inhibition of ROS production induced by 5-FU.
Agha-Hosseini et al. (2021) [66]	59	Triple-blind RCT	To evaluate whether a vitamin E, hyaluronic acid, and triamcinolone mouthwash was effective in the treatment of radiotherapy-induced OM grades 3–4.	Group with vitamin E + hyaluronic acid + triamcinolone rinses (IG); Group with triamcinolone rinses (CG). WHO scale	4 weeks	Significant reduction in the severity of RT-induced OM in IG over time.
Yeshurun et al. (2020) [67]	52	Multicentre double blind RCT	To determine whether folinic acid (FA) reduces methotrexate (MTX)-induced toxicity in patients undergoing myeloablative conditioning (CM) for allogeneic haematopoietic cell transplantation, who have also received MTX prophylaxis for graft-versus-host disease.	TG with FA; PG with placebo; WHO scale.	17 (4.5–50)	There were no significant TG and PG differences in incidence, duration, or severity of OM.
Pattanakitsakul et al. (2020) [68]	30	Preliminary and single-centre quasi-randomized trial	To examine the protective effect of vitamin A supplementation against mucosal damage of the gastrointestinal tract after CT in paediatric patients undergoing HSCT. As a secondary objective, to assess the occurrence of OM.	TG with single dose (200000 IU) of vitamin A; PG without vitamin A; WHO scale.	12	There were no significant differences in incidence or severity of OM.

## Data Availability

The study did not report any additional data.

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
