# Peer review of "A Narrative Review about Nutritional Management and Prevention of Oral Mucositis in Haematology and Oncology Cancer Patients Undergoing Antineoplastic Treatments"

_nutrients, 2021, doi:10.3390/nu13114075_

Round 1
Reviewer 1 Report
The study is a narrative review based on a systematic review of the literature, that aims to evaluate the effect of nutritional interventions on the prevention or treatment of oral mucositis in cancer patients undergoing radiotherapy and/or chemotherapy. The topic is very interesting. There are some issues mainly on the methodology of the study that need to be considered;
Introduction
- The authors stated and discussed all the mentioned interventions of the Clinical practice guidelines that have been discussed by the Multinational Association of Supportive Care in Cancer and the International Society of Oral Oncology (MASCC/ISOO) except the Photo-biomodulation (PBM), despite being recommended with a level of evidence II for the prevention of Oral mucositis (OM) in patients undergoing Hematopoietic Stem Cells Transplantation (HSCT). It is preferred to add a paragraph discussing the PBM application.
- References should be added for the part on page 2, lines 60-80
Methods
- The systematic approach was not described comprehensively. It is recommended to revise the methods section and develop it following the parameters of PRISMA guidelines. The search strategy is not clear, how did the authors search the PubMed database? What were the keywords and how were they combined?
- Some results of the electronic search were described in the section of data extraction, instead of describing how were the data extracted. It was understood that the selection of studies was carried out independently, was the extraction of data performed in the same way? In addition, in case of disagreement between the two reviewers in both of the two stages “selection of studies and extraction of data", how have the decision been taken? the first letters of the reviewers should be provided. A revision is needed with taking into consideration these issues.
Results
- In the section of Summary of the Studies Included, the authors did not give any references number for any paragraph. It is mandatory to add the references' numbers in this section (Page 5, line 169-204).
- A title of table 1 should be provided.
- In table 1, the studies were mentioned with the authors' names and year of publication. Also, the reference number should be added for each study.
Discussion
- The same issue of missing the references’ numbers is present. All the paragraphs should be provided with their references' numbers even that the authors' names of the studies are present.
- There are some limitations of this study that were noticed including; the possible presence of selection bias because of selecting only studies in English and Spanish languages, and this study was only carried out on one database. These limitations should be added or if the authors have justifications, they should be added to the manuscript.
Author Response
Dear reviewer,
We modified the manuscript as follows:
- Introduction
- We included some information about PBM.
- We added the references in page 2.
- Methods
- We described more comprehensively the systematic approach and the methods section.
- We added the keywords and search strategy.
- We modified the "Data Extraction" section.
- Results
- We added the references numbers.
- We added the title of table 1 and references numbers.
- Discussion
- We added the references numbers.
- We included and modified the possible presence of selection bias
Reviewer 2 Report
First need to use End Note or other program to get references in format of Nutrients. (if get Endnote, then need to download Nutrients - then references appear in the test as numbers inside brackets [1])
Second point - on glutamine. Although a narrative, this review may not help nutritionists and cancer patients know this is abeneficial nutrient for mucositis! - it just describes more research is needed.
1- Line 122 need to state" Oral glutamine plus disaccharide was shown to significantly help mucositis during chemotherapy and autologous BMT in randomized, placebo controlled trials (ref= ref Anderson, Schroeder, Skubitz, Oral Glutamine reduces duration and severity of oral mucositis after cytotoxic cancer chemotherapy Cancer (1998) 83:1433-1439) and Anderson, Ramsay et al. Bone Marrow Transplantation (1998) 22:339-344."
Line 286. Peter showed that Saforis was effective. Please delete" with no conclusive results to report". [FYI- results were conclusive This was an ASCO plenary session!]
Line 286 since Saforis is not commercially available. Add " a commercially available glutamine + trehalose powder to use as a suspension is now available". and cite Anderson and Lalla Nutrients 2020 reference.
In your conclusion of this paragraph (line 294), it would be more accurate to state "beneficial by topical absorption or ingestion if local uptake can be facilitated with a disaccharide. Although other studies have also show modest benefit of glutamine rinses or ingestion (citations), it would appear that achieving a local effect may possibly increase effectiveness. "
Author Response
Dear reviewer,
We modified the manuscript with your comments as follows:
- In the first time, we don't have any program to get references, but we modified the references with the numbers inside brackets.
- We have included the references you sent us (line 122, line 286), the Anderson and Lalla (2020) reference, and "more research is needed" about glutamine.
- We modified the line 294.
Thank you for your time and recommendations.
Round 2
Reviewer 2 Report
Revision is better. Encouraging to see the revised + improved version
Author Response
Dear Reviewer,
We have reviewed the translation and the language and style.
Thank you,